# Assessment and Valuation of Groundwater Ecosystem Services: A Case Study of Handan City, China

**Xuyang Yang [1,2,\*] and Jian Liu [2]**

[1]  School of Water Conservancy and Hydropower, Xi'an University of Technology, Xi'an 710048, China
[2]  Water Resources Research Institute of Shandong province, Jinan 250014, China; water_liujian@163.com
[\*]  Correspondence: 1180411017@stu.xaut.edu.cn

**Abstract:** Groundwater is important for maintaining ecosystem balance. However, the ecological value of groundwater is always undervalued, while its value as water resources is emphasized. Thus, this paper divided the systematic valuation of groundwater-dependent ecosystems (GDEs) into three categories, and its services into four categories based on the utilization. In addition, a service valuation system was developed, with ten indicators and nine assessment models for the GDEs. We then used this model to value the GDEs services of Handan city in 2015 as a case study, with seven relevant indicators. The results show that the total value of Handan's GDEs is 9.10 billion Renminbi (RMB, Chinese unit of currency), including a direct use value of 3.53 billion, and an indirect use value of RMB 5.57 billion. These results highlight the significant indirect use value of GDEs. Consequently, groundwater resources should be rationally exploited to maximize both direct and indirect service values on the basis of a comprehensive understanding of the inherent value of the resource.

**Keywords:** groundwater-dependent ecosystem; ecosystem classification; economic value estimate; indirect use value

## 1. Introduction

Groundwater is an important strategic resource for socioeconomic development and one of the important factors for ensuring ecological integrity and comfortable living environments [1]. In recent years, the overexploitation of groundwater and frequent human activities have led to a series of environmental issues, including land subsidence, desertification, groundwater pollution and the destruction of groundwater-dependent ecosystems (GDEs) [2]. The sole focus of groundwater overexploitation tends to be on threats to the direct use value of the resource, without sufficient knowledge of GDEs service values because the total GDEs service value has not been systematically analyzed [3]. A comprehensive analysis and assessment of GDE services is of great importance to groundwater conservation, because it facilitates the understanding of groundwater values and quantifies the economic and ecological values of groundwater.

The term 'ecosystem services' emerged in the early 1970s and has since become a scientific term [4]. In 1997, Daily et al. conceptualized ecosystem services in a more scientific manner, and defined them as the direct and indirect benefits of goods and services delivered by an ecosystem to ensure and enhance human well-being [5]. Constanza et al. divided global ecosystem services into 17 categories and monetized 10 types of biological communities, including oceans, bays, coral reefs, forests, grasslands, wetlands, lakes and rivers, deserts, farmlands and urban areas [6]. Scholars in China began to study ecosystem services in the 1990s, alongside increasing attention by ecologists worldwide, to quantify ecosystem services. For instance, Hou et al. systematically valued forest ecosystem services in their book 'Forest Resource Accounting in China' [7]. Xue determined the tourism and economic values of biodiversity in the Changbai Mountain Nature Reserve using economic

methods [8]. Ouyang et al. systematically elaborated the definitions, categories and assessment methods of ecosystem services [9], and preliminarily valued terrestrial ecosystem services in China [10]. As the primary factor ensuring the balance of various ecosystems, groundwater has developed into a new interdisciplinary field of study, and a service assessment system is now being established for various GDEs [11]. The majority of existing service assessments for GDEs involve functional analysis and valuations of groundwater-related ecosystems, such as wetlands, forests and swamps. However, the systematic analysis and valuation of groundwater ecosystem services in a monetized manner is rare [12]. For instance, Griebler et al. (2015) summarized the services delivered by groundwater ecosystem and qualitatively presented the service types of groundwater ecosystem without reporting a specific assessment method [13]. Furthermore, Shu et al. identified atmospheric regulations, water provisions, windbreaks and sand fixation and water and soil conservation as functional indicators of groundwater ecosystem services in the Erdos Desert Plateau, and valued those services based on each indicator [14]. Finally, Li reported a specific assessment method to value groundwater ecosystem by dividing them into 'deep resource functions' and 'shallow ecological functions' [15]. Hence, it is apparent that the systematic classification of groundwater ecosystem services and assessment methods are worth in-depth study. By considering the results of ecosystem services reported in previous studies, this study classifies GDEs and their service types, and analyses these services and their assessment methods. Furthermore, we determine the economic value of GDE services to provide a reliable foundation for establishing a scientific and comprehensive assessment system for GDE services.

## 2. GDE Classification Framework

Based on the definition of ecosystem services reported by Daily et al., we defined GDE services as the benefits of GDEs and ecological processes that contribute to ensuring sound human living environments and quality of life. GDE services not only provide fundamental products for human survival, but also support the environmental conditions which are essential for life on Earth [10]. Based on the provisional patterns of groundwater delivery to ecosystems, GDEs can be divided into three categories: groundwater-dependent ecosystems, ecosystems supported by groundwater overflows and ecosystems maintained by the groundwater level [12,16,17] (Table 1). Furthermore, we then classified GDE services into four categories according to the classification method described in the Millennium Ecosystem Assessment (MEA): provisioning, regulating, cultural and supporting services [18].

**Table 1.** Classification and service sub-types.

| Type | Sub-Type | Features | Service Types | Service Indicators |
|---|---|---|---|---|
| Groundwater ecosystems | Unconfined aquifer and karst cave ecosystems | Aquifer serves as water resources (including abstracted water for human activities); aquifer pores, cracks and karst caves contain special minerals, and serves as habitats for invertebrate aquatic organisms. | Provisioning | Water supply (domestic, industrial, agricultural and power generation activities) |
| | | | Regulating | Water conservation |
| | | | | Air regulation |
| | | | | Water purification |
| | | | | Organic matter deposition |
| | | | | Elimination of pathogens |
| | | | Cultural | Scientific research |
| Ecosystems supported by groundwater overflows | River ecosystem Wetland ecosystems Bay ecosystems | a. Aquatic habitats in rivers b. Aquatic ecosystems, such as swamp, fountains, artificial wetlands and lakes c. Ecosystems, such as river bays, shallow sea and beach | Provisioning | Food, raw materials, water resources, etc. |
| | | | Regulating | Flood control |
| | | | | Air regulation |
| | | | | Water purification |
| | | | | Soil conservation |
| | | | | Climate regulation |
| | | | Cultural | Scientific research |
| | | | | Recreation |

**Table 1.** *Cont.*

| Type | Sub-Type | Features | Service Types | Service Indicators |
|---|---|---|---|---|
| Ecosystems maintained by the groundwater level | Terrestrial vegetation ecosystems Near-shore vegetation ecosystem. | a. Unconfined aquifer directly supplies water to vegetation roots b. Near-shore vegetation in rivers and wetlands relies on groundwater during the dry season | Provisioning | Food and raw materials |
| | | | Regulating | Flood control |
| | | | | Air regulation |
| | | | | Waste treatment |
| | | | | Soil conservation |
| | | | | Pollination |
| | | | | Nutrient cycle |
| | | | | Climate regulation |
| | | | Cultural | Scientific research |
| | | | | Recreation |
| | | | Supporting | Habitats |

This paper analyzed and valued the groundwater-dependent ecosystem services. Provisioning services of groundwater-dependent ecosystems refers to the water resources delivered by the unconfined aquifer for agricultural, industrial, domestic and energy generation activities [19]; regulating services refers to benefits delivered by the evolution and accumulation of groundwater, including water conservation, water purification, air regulation, organic matter deposition and the elimination of pathogens [20]; cultural services refers to the scientific value of endemic invertebrates, microorganisms and living fossils in groundwater.

## 3. Service Valuation Methods

In general, there are two ecosystem service valuation methods: the surrogate market technique and the simulated market technique (also known as the hypothetical market technique). The surrogate market technique expresses the economic value of ecosystem services using 'shadow prices' and consumer surplus, involving various assessment methods, including market value (productivity), surrogate, travel cost, restoration cost, opportunity cost and shadow project approaches. The simulated market technique expresses the economic value of ecosystem services using willingness to pay and the net willingness to pay, involving only one assessment method, the contingent valuation method [21]. Here, we selected suitable methods to establish an indicator system for assessing the groundwater ecosystem services, as shown in Table 2.

**Table 2.** Service valuation system for groundwater ecosystems.

| Service Type | Service Indicators | Assessment Type | Assessment Method |
|---|---|---|---|
| Provisioning | Water supply (domestic, industrial, agricultural and energy generation activities) | Direct | Market value approach |
| Regulating | Water conservation | Indirect | Shadow project approach |
| | Air regulation | | |
| | Water purification | | Replacement cost approach |
| | Organic matter deposition | | Surrogate approach |
| | Elimination of pathogens | | |
| Cultural | Research culture | Direct | Travel cost approach |

### 3.1. Valuation Methods for Provisioning Service

#### 3.1.1. Value of Groundwater for Agricultural Use

Agricultural planting areas can be divided into irrigated areas and non-irrigated areas (dry lands). The irrigation value of groundwater was estimated by setting the producer surplus as the price that farmers were willing to pay for water. The producer surplus was estimated according to the fixed costs of water used in irrigated areas. Gross profits refer to the difference between gross income and production costs; in other words, the short-term groundwater value in agricultural irrigation approximates the producer surplus over a certain period of time, and thus can be estimated according to the agricultural profits of irrigated and non-irrigated areas. In this review, the typical agricultural products of an area were selected based to the model constructed by Kulshreshtha (1994) to estimate the value of groundwater in agricultural irrigation over a certain period of time [22], using the market value approach via the following equation:

$$V_1 = A_1 \times [(P \times YD_i - C_i) - (P \times YD_d - C_d)] = A_1 \times [P \times YD_i - YD_d) - (C_i - C_d)] \tag{1}$$

where $V_1$ is the value of groundwater for agricultural use, $P$ is the price of agricultural products at a given time, $A_1$ is the area of agricultural under production at that time, $YD$ is the agricultural output per unit area ($i$ refers to irrigated areas while $d$ refers to dry lands) and $C$ is the annual investment cost (in this study based on 2015 data).

#### 3.1.2. Value of Groundwater for Industrial Use

The economic value of groundwater in industry was estimated using consumer surplus as the willingness to pay the water bill. A key parameter of consumer surplus assessment is the elasticity and magnitude of the water demand. On the basis of previous studies, we investigated the elasticity of industrial demand for water. According to the model constructed by Muller (1985) [23], the industrial value of groundwater approximates the average willingness to pay or the value of consumer surplus. Therefore, the valuation equation of groundwater for industrial use is as follows:

$$V_2 = P_0 Q_1 \frac{[P_a \div P_0]^{n+1} - 1}{n+1} \tag{2}$$

where $V_2$ is the value of groundwater for industrial use, $P_0$ is the benchmark price of water used in industry, $P_a$ is the affordable price of water used in industry, $Q_1$ is the consumption of water in industry (groundwater) and $n$ is the elasticity of industrial demand for water within the given area.

#### 3.1.3. The Value of Groundwater for Domestic Use

The value of groundwater for domestic use was estimated using the same valuation equation as the groundwater for industrial production. However, the actual definition of each parameter is different between the two equations. The valuation equation of groundwater for domestic use is as follows:

$$V_3 = P_1 Q_2 \frac{[P_b \div P_1]^{n+1} - 1}{n+1} \tag{3}$$

where $V_3$ is the value of groundwater for domestic use, $P_1$ is the price of urban domestic water, $P_b$ is the affordable price of urban domestic water, $Q_2$ is the consumption of urban domestic water (groundwater) and $n$ is the elasticity of demand for urban domestic water.

### 3.1.4. The Value of Groundwater in Geothermal Power Generation

Subterranean heat is retrieved using deep underground facilities and converted to electricity via geothermal power plants or used directly for heating purposes [13]. The economic value of geothermal energy was estimated via the market value approach using the following equation:

$$V_4 = W_1 \times C_1 \tag{4}$$

where $V_4$ is the value of groundwater in geothermal power generation, $W_1$ is the total geothermal power generation and $C_1$ is the per unit price of electricity for geothermal power generation.

### 3.2. Regulating Service Valuation Methods

#### 3.2.1. Water Conservation

Groundwater resources refers to the amount of underground freshwater that can be provided for human use within a certain period of time and can be recovered year by year. Groundwater reserves are abundant water resources that can be used to recharge and regulate river runoff, lake water and for water conservation. The economic value of water conservation can be estimated via a shadow project approach, to simulate the reservoir cost within the given area, using the following equation:

$$V_5 = Q_3 \times C_2 \tag{5}$$

where $V_5$ is the value of groundwater in water conservation, $Q_3$ is the amount of groundwater resources within the given area and $C_2$ is the reservoir cost.

#### 3.2.2. Water Purification

Water purification refers to the elimination of external pollutants from the groundwater within an aquifer through mechanisms, such as soil exchange and adsorption, biodegradation etc., in the case of good groundwater quality. The value of purified groundwater can be estimated via the cost of sewage treatment using the following equation:

$$V_6 = Q_4 \times f_1 \times C_3 \tag{6}$$

where $V_6$ is the value of groundwater in water purification, $Q_4$ is the amount of sewage discharged per year, $f_1$ is the pipeline leakage rate and $C_3$ is the sewage treatment cost for that period.

#### 3.2.3. Air Regulation

Human domestic and industrial activities lead to increasing organic matter content in groundwater that results in the accumulation of vast quantities of carbon in groundwater. Furthermore, groundwater also generates $CO_2$ emissions due to biochemical reactions in the soil and various qualitative changes within the earth's crust. In other words, the exploitation of ground water releases $CO_2$ equivalent to the combustion of approximately 9000 tons of pure coal per 100 million $m^3$ of groundwater, based on its average total carbon content [24]. The value of groundwater ecosystems in carbon fixation can be calculated by afforestation costs via a shadow project approach using the following equation:

$$V_7 = Q_5 \times C_4 \tag{7}$$

where $V_7$ is the value of groundwater in air regulation, $Q_5$ is the total amount of fixed carbon in the groundwater within the given area and $C_4$ is the regional afforestation costs.

### 3.2.4. Nutrient Deposition

Groundwater nutrients mainly refer to sodium, potassium, phosphorus and nitrogen ions. Terrestrial vegetation that relies on groundwater and other ecosystems that rely on groundwater discharges, such as surface water, wetlands and swamps, can extract nutrients from groundwater. The value of this service can be estimated according to the average price of chemical fertilizers using the following equation:

$$V_8 = W \times k \times C_5 \tag{8}$$

where $V_8$ is the value of nutrients, $W$ is the total nutrient content of groundwater, $k$ is the relative content of each ion and $C_5$ is the average price of chemical fertilizers.

### 3.2.5. Elimination of Pathogens

Microorganisms found in groundwater ecosystems can potentially attenuate, inactivate, and eliminate pathogens. More than 20% of 217 pure bacterial isolates from groundwater were found to inhibit the growth of *Escherichia coli* K12 strain [13]. Unfortunately, the economic value of this service remains elusive and, without a comprehensive assessment model, it cannot be fully exploited at this time.

### *3.3. Cultural Service Valuation Methods*

Scientific Research

Groundwater ecosystems are rich in living fossils and endemic species with a high scientific value, which can be estimated via the expenditure on groundwater-related research projects using the following equation [25].

$$V_9 = Z \times C_6 \tag{9}$$

where $V_9$ is the scientific value, $Z$ is the number of groundwater-related research projects and $C_6$ is the average expenditure of each research project during 2015.

## 4. Case Study

Handan City is located at the southernmost margins of Hebei Province with latitude ranges from 36°04′–37°01′ N and longitude ranges from 113°28′–115°28′ E (Figure 1). According to the statistics of Handan Statistical Yearbook, as of 2015, the total population of Handan is 9.4 million, and the population density is 782 people/km$^2$. The city is a resource-poor area with a humid continental climate, having summer rainfall and dry winters. In 2015, the city had annual precipitation of 412.0 mm and 872 million m$^3$ of groundwater, which were 136.9 mm and 396 million m$^3$ less than their respective average annual values. According to the water quality classification of "Groundwater Quality Standards (GB/T14848-93), the groundwater quality stations in Handan are mostly rated V and IV, and may continue to deteriorate. In addition to the influence of primary hydrogeological factors, the groundwater quality in Handan is mainly related to the long-term over-exploitation of groundwater [26–28]. Since the beginning of the 21st century, Handan City has had to rely heavily on groundwater overexploitation for domestic, industrial and agricultural uses to support the rapid economic development of the region, resulting in land subsidence, well abandonment, groundwater pollution and building destruction.

In this study, we have valued the provisioning services (water supply for agricultural, industrial and domestic uses), regulating services (water conservation, water purification and air regulation), and cultural services (scientific research) of groundwater based on data collection from groundwater ecosystems in Handan City. Among the seven services assessed in this study, the direct use values include groundwater for agricultural, industrial, domestic and scientific research uses, while indirect use values include groundwater for water conservation, water purification and air regulation. These data will provide a reference for the valuation of groundwater ecosystem services in other plain regions

of Northern China. Estimates used in the valuation calculations were based on data from the '2015 Handan City Water Resources Bulletin' and 'Handan Statistical Yearbook 2015'. Estimates of ecosystem service values relate to services provided for the 2015 calendar year only.

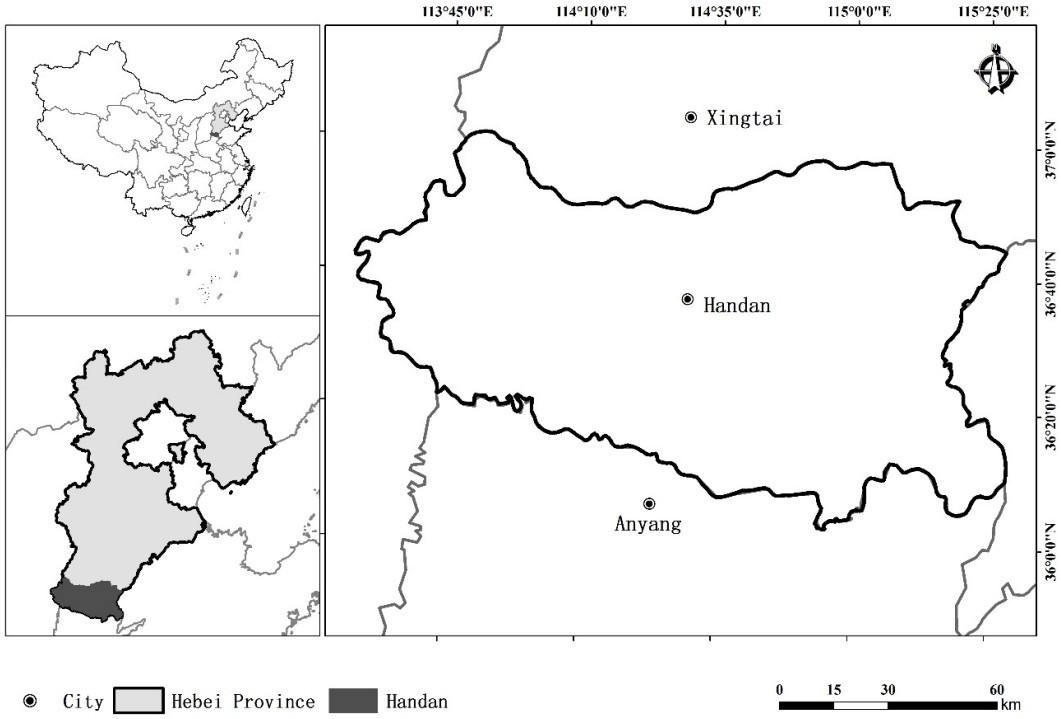

**Figure 1.** The location map of the Handan Area.

*4.1. Provisioning Service Valuation*

4.1.1. Value of Groundwater for Agricultural Use

In 2015, Handan City had approximately 8.07 million acres of irrigated agricultural land. Winter wheat and summer maize were selected as the typical agricultural products of the region, because they are both widely cultivated. The survey parameters of typical agricultural products in Hebei Province are shown in Table 3. In this study, the prices of wheat and maize were retrieved from the official website of the National Development and Reform Commission, while their input costs (excluding the labor cost) and yields were retrieved and averaged from previous studies [29,30] and the '2015 National Agricultural Product Cost and Revenue Information'. According to the calculation of Equation (1), the value of groundwater in agricultural production is RMB 3.219 billion.

**Table 3.** List of parameters of typical agricultural products in Hebei Province.

| Price/Yield/Cost | Irrigated Areas | Drylands |
|---|---|---|
| Wheat market price/(RMB/kg) | 2.36 | 2.36 |
| Maize market price/(RMB/kg) | 2.00 | 2.00 |
| Wheat yield/(kg/hm$^2$) [1] | 6885.00 | 5086.50 |
| Maize yield/(kg/hm$^2$) | 7282.50 | 5952.00 |
| Wheat input cost/(RMB/hm$^2$) | 3262.50 | 2730.80 |
| Maize input cost/(RMB/hm$^2$) | 1828.50 | 1440.90 |

[1] hm$^2$ refer to square hectometer.

### 4.1.2. Value of Groundwater for Industrial Use

Handan City consumed 238 million m$^3$ of water for industrial use in 2015, including 140.8 million m$^3$ of groundwater. Based on the data from China water price website [31], the benchmark price of water for industrial use in Handan City is RMB 4.85 per m$^3$. Using data retrieved from previous studies and the current development status of Handan City, the selected demand elasticity and affordable price of water for industrial use were set at −0.163 and RMB 5.86/m$^3$, respectively [32,33]. Therefore, the value of groundwater for industrial use in Handan City, calculated by Equation (2), is RMB 140 million.

### 4.1.3. Value of Groundwater for Domestic Use

Handan City consumed 149 million m$^3$ of water for urban domestic use in 2015, including 98 million m$^3$ of groundwater. Based on the data from China water price website [31], the total urban domestic water price in Handan City was RMB 3.55/m$^3$ (before the implementation of a progressive pricing scheme for water use). Water consumption is negatively correlated to water price. Therefore, water consumption drops by approximately 28% as the water price doubles [34]. Based on retrieved data and the current development status of Handan City, the selected demand elasticity and affordable price of urban domestic water in this study were set at −0.144 and RMB 5.33/m$^3$, respectively [35,36]. Hence, the value of groundwater for domestic use in Handan City, calculated by Equation (3), is RMB 168 million.

### 4.2. Regulating Service Valuation

### 4.2.1. The Value of Groundwater for Water Conservation

Handan City had 872 million cubic meters of groundwater in 2015 and the average reservoir cost in Hebei Province was approximately RMB 4.13 per cubic meter [15]. Therefore, the value of groundwater for water conservation in Handan City, calculated by Equation (5), is RMB 3.601 billion.

### 4.2.2. The Value of Groundwater for Water Purification

Groundwater exhibits a certain self-purification ability without compromising its own quality. Based on the survey data, we only considered the purification ability of groundwater against nonpoint source (NPS) pollution. In 2015, the total wastewater and sewage discharges of Handan City were 108,785,000 m$^3$, including 38,650,000 m$^3$ of industrial wastewater discharge and 70,130,000 m$^3$ of domestic wastewater discharge, which accounted for 35.5% and 64.5% of the total discharge, respectively. Considering the 'Municipal Sewerage Quality Inspection and Evaluation Standards CJJ3-90' [37] and the basic conditions of the pipeline network in Handan City, the annual amount of sewage leakage into the ground was 435.14 million m$^3$, assuming that the pipeline network leaked 4% of the total sewage discharge. According to China water price website [31], sewage treatment in Handan City costs RMB 1/m$^3$ and thus, the value of groundwater for water purification services in Handan City, calculated by Equation (6), is RMB 4 million.

### 4.2.3. Value of Groundwater for Air Regulation

The amount of fixed $CO_2$ in the region's groundwater is 1.49 million tons, calculated from the amount of $CO_2$ (1.9 tons) released from the combustion of one ton of pure coal (available online: www.tanpaifang.com), and the volume of groundwater in Handan City (872 million m$^3$). Therefore, the value of groundwater in Handan City for air regulation, calculated by Equation (7), is RMB 1.967 billion (assuming that the $CO_2$ afforestation cost is RMB 1320/ton C [38]).

### 4.3. Cultural Service Valuation

The National Knowledge Infrastructure (CNKI) and Science Direct databases listed a total of nine Handan groundwater-related research articles published in 2015. Based on the research outcomes of Wang et al., with respect to the scientific value of marine ecosystems in the Yellow Sea [39], the average expenditure for each research project is approximately RMB 357,600 and, thus, the scientific value of groundwater ecosystems in Handan City is RMB 3 million.

### 4.4. Analysis of the Assessment Results

GDE service values in Handan City are summarized in Table 4. The total GDE service value in Handan City was RMB 9.102 billion, equivalent to 3% of the GDP of Handan City in 2015. These include the values of providing, regulating and cultural services, which were RMB 3.527 billion, RMB 5.572 billion and RMB 3 million, respectively. In 2015, GDE service values in Handan City were mainly attributed to water for agricultural use (RMB 3.219 billion), water conservation (RMB 3.601 billion) and air regulation (RMB 1.967 billion), which accounted for 95.3% of the total value. In 2015, the service value of 100 million m$^3$ of groundwater in Handan City was approximately RMB 1.044 billion.

**Table 4.** Service values in Handan City estimated for the year 2015.

| Service Types | Service Indicators | Value/RMB 100 Million | Total |
|---|---|---|---|
| Provisioning | Agriculture | 31.05 | 35.27 |
| | Industry | 1.4 | |
| | Domestic | 1.68 | |
| Regulating | Water conservation | 36.01 | 55.72 |
| | Water purification | 0.04 | |
| | Air regulation | 19.67 | |
| Cultural | Scientific research | 0.03 | 0.03 |
| Total | | 91.02 | 91.02 |

## 5. Conclusions

GDE services were categorized into provisioning, regulating, cultural and supporting services according to the MEA classification scheme. Additionally, a preliminary assessment system was established, consisting of three service types, ten service indicators and nine assessment models, on the basis of groundwater ecosystem structures and utilities. Seven of the service indicators used the surrogate market technique (market value, shadow project and travel cost approaches). We conducted a preliminary valuation of the groundwater ecosystem services, and the seven service indicators of the three service types in Handan City. However, the estimated values were relatively small, due to some uncertainties during the assessment. Firstly, a geothermal power plant has not been constructed in Handan City, and the city has forbidden the use of groundwater for thermal energy to protect the groundwater resources. Secondly, the volume and composition of nutrients in groundwater supplied to terrestrial vegetation, surface water and wetlands in the Handan area could not be estimated accurately, due to data scarcity. Excessive nutrients may contaminate the groundwater and lead to eutrophication of surface waters and wetlands. Therefore, the value of this service indicator needs to reflect its value based on the quality of groundwater. Finally, the function of bacteria isolated from the groundwater in inhibiting the growth of other pathogens cannot be estimated at present.

Our results clearly show that our assessment still requires further study and improvement, due to the incomplete set of indicators and quantitative estimation. However, our results clearly that GDEs provided significant ecosystem services to the value of RMB 9.102 billion in 2015. The indirect use value was 1.57 times greater, as compared to the direct use value, highlighting the significant value of indirect use. The long-term over-exploitation of groundwater in the study area has caused a series of ecological and environmental problems, which have seriously affected the social and economic

development of the region and the well-being of the people. China vigorously promotes ecologically sustainable development. According to the deployment and requirements of China's groundwater over-exploitation and governance, the local government has gradually carried out the management of groundwater over-exploitation, and strives to gradually rationalize groundwater development to a balanced state. Due to the lack of data, this article failed to assess the loss of GDE service values by excessive groundwater exploitation. The exclusive pursuit of the direct economic values of groundwater will inevitably lead to overexploitation and water pollution that compromises indirect service values. Under these circumstances, the distribution and balance of ecosystems that rely on groundwater, such as wetlands and swamps, will be greatly altered. Although the overexploitation of groundwater can increase its direct use value, in the long run, the quantity and quality of groundwater will continue to deteriorate. When there is no groundwater available, how can we talk about GDE service values? Hence, an appreciation of GDE services and their values are of high practical significance for rapid, yet sustainable, socioeconomic development. This study provides a foundation for policymakers and decision-makers to attach importance to groundwater resources and ecosystems, and proposes strategic advice on the rational exploitation and utilization of groundwater.

**Author Contributions:** Resources, X.Y.; data curation, X.Y.; writing—original draft preparation, X.Y.; writing—review and editing, J.L. All authors have read and agreed to the published version of the manuscript.

**Funding:** This research received no external funding.

**Acknowledgments:** This work was supported by the Science and Technology Project of Hebei Province (Grant No. 15227005D) and Scientific Research Project of Hebei Provincial Education Department (QN2016233, ZD2016131).

**Conflicts of Interest:** The authors declare no conflict of interest.

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
