# Peer review of "Assessment and Valuation of Groundwater Ecosystem Services: A Case Study of Handan City, China"

_water, doi:10.3390/w12051455_

Round 1
Reviewer 1 Report
This is a short, interesting paper that attempts to quantify the value of ecosystem services (ES)provided by GDEs. The paper uses several approaches to quantify different service values. The approaches taken are basic, and for many of the indicators, the reader is expected to accept some very large caveats. Nevertheless, the paper highlights the immense value of ecosystem services.
The point about this paper that I struggle to reconcile is that the paper predicts the value of services under the current conditions of heavy groundwater exploitation. It would seem that the high economic value of the ES is a direct consequence of this over exploitation (particularly since a large value is derived from the use of groundwater for irrigation). In a way, this paper highlights the values and benefits of overexploitation of the resource rather than being a voice for conservation and sustainability.
I would like to see in this paper some estimate of the services provided if the groundwater resource was managed sustainably. What is the difference in the ES between sustainable GW abstraction and management, and the current state of high rates of use and contamination? Some mention of when ES will be lost by overexploitation is a strong (and much needed) message to send.
The paper is nicely written and logically constructed. It is well references, although the authors might like to consider Murray et al 2006. Valuation of groundwater dependent ecosystems: a functional methodology incorporating ecosystem services. Australian Journal of Botany. 54: 221-229 that discusses approaches such as those taken here in the context of GDEs.
Throughout the paper, I did not get a feel for the timing of the value estimates. Please make it clear if the estimated values are per year, or if they involve some forward or backward casting. If they are values for the current year, it would be useful to have some discussion about likely future changes, including climate change impacts into the future.
Specific comments
L157 generates CO2 emissions
L216, 217 this is a very large value – is this an annual figure or does it involve some future and back casting?
Table 3 what is hm2?
Author Response
请参阅附件。

Reviewer 2 Report
Dear Authors,
The manuscript entitled: "Assessment and valuation of groundwater ecosystem
3 services: A case study of Handan City, China" tackle an important issue.
The manuscript need to be developed especially in the introduction and reference parts, in addition to some other points.
Please find the attached pdf file which includes all of my comments.
Best wishes

Author Response
I have made modifications to all your comments in the corresponding parts of the article.
Thank you for your time and condideration. During the period of the epidemic. I hope you can stay on guard and hope all is well with you and your family.
Reviewer 3 Report
This manuscript defines and addresses a subject of interest in a world where water resources are becoming scarce. My comments largely address issues of clarity, rather than the substance of the research.
As a broad comment, when "billions" of something are mentioned, two decimal places is adequate.
Line 6: Shandong should be capitalized.
Line 13: Please spell out RMB once, here or in the Introduction. I know it’s currency but not exactly what.
Lines 18-19: Keywords should be terms not in the title, because the terms are already searchable as part of the title. I strongly suggest dropping ecosystem services, and Handan, from this list and find others.
Line 35 and later: The Reference citation numbers beginning here are both out of sequence, and missing the correct source in the Reference list. Because at least one citation is missing in the list all of them are confused in some irregular fashion.
Line 39-44: The sentence beginning with, “For instance, Hou et al. …” is a very long run-on sentence and should be broken into two or three sentences.
Line 74: Should be, “…refers to the water resources…”
Line 76: Should be, “refers”.
Line 78: Should be, “refers”.
Line 91, Table 1: In the header, should be “Sub-types”. Under Type, the first category has a bad word break, keep Groundwater together. Under the first Features, third line from bottom should read, “serves as habitats for”. Under Service types, all references to services in that column should be eliminated as redundant because the title already describes them as such.
Line 92, Table 2: In column titles, it should be, “Service types” Within the Service types column, again the word “services” is redundant and should be eliminated.
Line 139: “Groundwater resources” should be briefly defined. I know broadly what you mean, but specifically which local groundwater resources are you analyzing?
Line 157: CO2 is in the middle of a run-on word block and needs word spaces on either side.
Line 169: This sentence mentions that various water bodies can “retrieve nutrients from groundwater.” Does this mean these bodies work as purifiers and remove excess nutrients? It may be simply that “retrieve” is the wrong word to use here.
Line 178: “Escherichia coli” should be in italics.
Line 194: “m3less” is really two words, and needs a word space.
Line 230: “consumption reduces” should be “consumption drops”?
Line 235, Table: In the column heads, “Irrigated areas” and “Drylands” need units of measure listed with them.
Line 243: This sentence does not make any sense, needs to be re-written.
Line 248: The “Municipal Sewerage Quality Inspection and Evaluation Standards CJ3-90” needs a citation.
Line 261: Citations are needed for the CNKI and Science Direct databases, somehow. Plus, reference 32 is wrong and out of sequence, as are most other citations…..
Line 268-269: Instead of beginning a sentence, “Among which,”, it reads better as, “These include”.
Line 270: “Handan City were mainly”, not “was”.
Line 274, Table 4: In the column, Service types, the word services is again redundant and should eliminated in the three blocks.
Line 276: The sentence should begin, “GDE services were”, and eliminated “Furthermore,”.
Line 284: Is using the word “abstraction” here accurate? Should the term be “use”?
Line 290: Rather than use “medicinal value”, simply using “function” reads better.
Line 293: “our results clearly show that”, insert show.
Line 294: Instead of “in 2015, that the”, it should be “in 2015, and the”.
Line 350: Along with the References being jumbled in general, there is no 20 in the sequence.
Author Response
I have made modifications to all your comments in the corresponding parts of the article.
Thank you for your time and consideration. During the period of the epidemic. I hope you can stay on guard and hope all is well with you and your family.
Round 2
Reviewer 1 Report
I appreciate the changes made to improve the MS, particularly the additional discussion around the over exploitation of the resources and the political context of such discussions and the critique of government policy.
I am still not satisfied that the explanation of the timing of the cost estimates is clear. Although the response highlights that the data are for 2015, there is no explicit statement that says the values are estimated for that calendar year. Throughout the MS, this needs to be made clear. I suggest including a statement at line 215 saying. Something like ‘Estimates of ecosystem service values relate to services provided for the 2015 calendar year only.’ As a further example, the caption of Table 4 should be expanded to state ‘GDE service values in Handan City estimated for the year 2015.’ L191 could be expanded to say ‘…expenditure of each research project during 2015.’
Specific comments
L8 suggest change ‘being’ to ‘is’
L26, 205 delete ‘quality’ (groundwater pollution)
L47-50 Authors may wish to include mention of Murray BR, Hose GC, Lacari D, Eamus D. 2006. Valuation of groundwater dependent ecosystems: a functional methodology incorporating ecosystem services. Australian Journal of Botany. 54: 221-229 https://doi.org/10.1071/BT05018
It seems that his paper is directly relevant to this study
L92 Table 1 It would be useful to provide a reference for the GDE classification scheme used (see Eamus et al 2006 which is already cited in the MS) – this should be mentioned in the table. Typically cave and aquifer ecosystems are referred to as just groundwater ecosystems so I suggest changing ‘groundwater dependent ecosystems’ in row 2 to ‘groundwater ecosystems’
L107-9 suggest change to ‘..price of agricultural products at a given time, A1 is the area of agricultural under production at that time, YD is the agricultural output per unit area (…..), and C is the annual investment cost (in this study based on 2015 data).’
L155-6 suggest change to ‘…amount of sewage discharged per year, f1 is the pipeline leakage rate, and C3 is the sewage treatment cost for that period.’
L160, 162 CO2 – fix subscript
L161 – over what time frame is this CO2 released? How does that value relate to the annual cost for 2015?
L197 people/km2
L199-200 please explain what V and IV are – they are not mentioned previously in the MS
L193-205. Please reorder sentences in this paragraph so that sentences relating to climate are together
L231,239 insert ‘the’ - ‘data from the China water….’
L307 suggest change to ‘.. our results clearly show that GDEs provided significant ecosystem services to the value of RMB 9.102 billion in 2015. The indirect use value was……’
L312-13 suggest change to ‘China vigorously promotes ecologically sustainable development. According to the…’
L321-23 I appreciate the inclusion of this important statement.
L323 When (capital W)
Author Response
According to your comments, I have made changes in the corresponding positions of the article one by one.
Thank you for your comments and suggestions. Good health to you.
Reviewer 2 Report
I think the manuscript was enhanced significantly. I think that the manuscript is suitable for publication in Water journal with the current form.
Author Response
想您的意见和建议。
身体健康。
This manuscript is a resubmission of an earlier submission. The following is a list of the peer review reports and author responses from that submission.